# TreeRPO: Tree Relative Policy Optimization

## Abstract

Large Language Models (LLMs) have shown remarkable reasoning capabilities through Reinforcement Learning with Verifiable Rewards (RLVR) methods. However, a key limitation of existing approaches is that rewards defined at the full trajectory level provide insufficient guidance for optimizing the intermediate steps of a reasoning process. To address this, we introduce **TreeRPO**, a novel method that estimates the mathematical expectations of rewards at various reasoning steps using tree sampling. Unlike prior methods that rely on a separate step reward model, TreeRPO directly estimates these rewards through this sampling process. Building on the group-relative reward training mechanism of GRPO, TreeRPO innovatively computes rewards based on step-level groups generated during tree sampling. This advancement allows TreeRPO to produce fine-grained and dense reward signals, significantly enhancing the learning process and overall performance of LLMs. Experimental results demonstrate that our TreeRPO algorithm substantially improves the average Pass@1 accuracy of Qwen-2.5-Math on test benchmarks, increasing it from 19.0% to 35.5%. Furthermore, TreeRPO significantly outperforms GRPO by 2.9% in performance while simultaneously reducing the average response length by 18.1%, showcasing its effectiveness and efficiency.

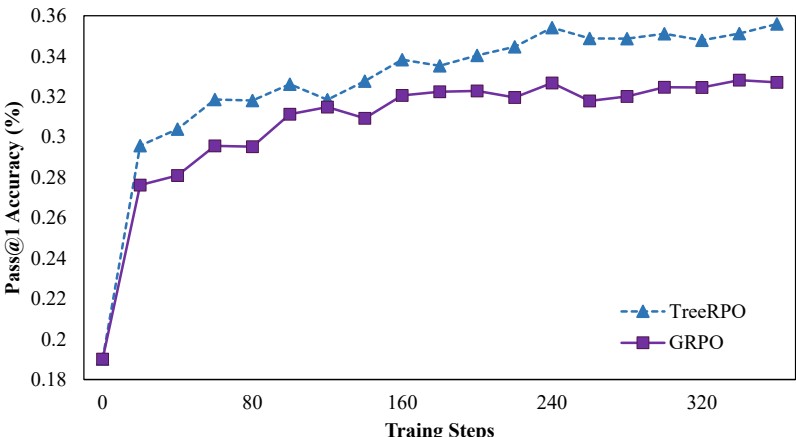

Figure 1: The average Pass@1 accuracy of TreeRPO and GRPO with Qwen-2.5-Math-1.5b on four mathematical benchmarks: MATH-500, OlympiadBench, Minerva, and AIME.

## 1 Introduction

Recent advancements in test-time scaling with reinforcement learning methods bring milestone progress to Large Language Models (LLMs). Reasoning models such as OpenAI O1 (OpenAI, 2024), DeepSeek R1 (Guo et al., 2025), and QwQ (Qwen, 2024) have demonstrated significantly superior performance in complex reasoning tasks. Reinforcement Learning with Verifiable Rewards (RLVR) plays a pivotal role in this progress, which enhances the model's performance by continuously exploring reasoning paths toward correct answers on verifiable problems.

In the realm of LLM-RL integration for complex reasoning, existing approaches can be broadly categorized into two paradigms: *reward model-based* methods (Ouyang et al., 2022; Shen et al.,

2025; Schulman et al., 2017) and *reward model-free* methods (Rafailov et al., 2023; Shao et al., 2024; Zeng et al., 2025; Luo et al., 2025b). Among reward model-based methods, reward models are typically divided into outcome reward models (ORMs; Cobbe et al. 2021; Yu et al 2023) and process reward models (PRMs; Lightman et al. 2023; Wang et al. 2024; Lu et al. 2024a; Chen et al. 2025). ORMs provide a single scalar reward for the entire generation sequence, while PRMs offer step-wise evaluations of the reasoning path. The fine-grained, dense reward signals from PRMs generally yield superior RL performance compared to ORMs. However, acquiring high-quality training data for PRMs remains challenging, as accurately annotating the correctness of intermediate reasoning steps requires substantial domain expertise. This data scarcity significantly hinders the scalability of PRM-based approaches.

Recent breakthroughs in enhancing LLM reasoning capabilities, such as GRPO (Shao et al., 2024) and its variants (Yu et al., 2025; Yue et al., 2025), have adopted a reward model-free paradigm. These methods leverage verifiable reward functions trained on complex reasoning datasets, where rewards are determined by whether the model's final output matches the ground-truth numerical answer or passes predefined unit tests in programming tasks. This approach achieves remarkable scalability by eliminating the need for human annotations or reward models. However, similar to ORMs, these rule-based methods only provide trajectory-level rewards, offering limited guidance for optimizing intermediate reasoning steps. Consequently, the question of how to deliver dense, fine-grained reward signals without relying on reward models presents an important research direction.

To address this challenge, we propose **TREERPO**, a novel approach that estimates the mathematical expectations of rewards at various reasoning steps through tree sampling. Unlike previous methods that require explicit step-level reward models, TREERPO employs a tree-based sampling mechanism to approximate these expectations. Building upon GRPO's group-relative reward training framework, TREERPO innovatively computes rewards based on step-level groups within the sampled tree structure. This design enables the generation of fine-grained, dense reward signals that guide the model's reasoning process more effectively while maintaining the scalability advantages of verifiable reward functions. Through this approach, TREERPO achieves more efficient and effective optimization of LLM reasoning capabilities.

To summarize, our main contributions are as follows:

- To the best of our knowledge, TREERPO is the first reward model-free method that provides dense process reward signals through tree sampling and group relative reward computation, significantly enhancing the efficiency of RL-based reasoning optimization.

- Through extensive experimentation, TREERPO was found to significantly increase Qwen-2.5-Math-1.5B's Pass@1 accuracy on various benchmarks from 19.0% to 35.5%, including a **2.9%** lead over GRPO.

- Detailed analysis demonstrates that TREERPO achieves higher accuracy and reduces token consumption. Specifically, it cuts the average response length on test benchmarks by **18.1%** compared to GRPO, showcasing more efficient and precise reasoning.

## 2  RELATED WORKS

### 2.1  ELICITING COMPLEX REASONING ABILITY

Complex reasoning tasks (Hendrycks et al., 2021; He et al., 2024; Lewkowycz et al., 2022; Zeng et al., 2024; Yang et al., 2025; Xiang et al., 2025) such as mathematical problem solving are one of the most challenging tasks for LLMs. Various methods are proposed to elicit the reasoning ability of LLMs. These approaches can be divided into two groups:
1) *In-context learning*: These methods aim to improve the reasoning ability of LLMs by designing various prompting strategies and frameworks without updating the model parameters. Chain-of-thought (CoT; Wei et al. 2022) prompting shows that intermediate reasoning steps can greatly improve model performance. Subsequent research (Zhang et al., 2023; Yao et al., 2023; Bi et al., 2023; Yang et al., 2024b) has further enhanced CoT through various methods.
2) *Fine-tuning*: This line of approaches (Yang et al., 2022; Yu et al., 2024; Lu et al., 2024b; Huang et al., 2024; Tong et al., 2024) involve finetuning on extensive and high-quality datasets to improve reasoning capabilities. The core of these methods is to construct high-quality question-response

pairs with chain-of-thought reasoning processes. MetaMath (Yu et al., 2024) focuses on data augmentation for both questions and answer texts. MathGenie (Lu et al., 2024b) collects a vast amount of data through open-source language models. DART-Math (Tong et al., 2024) generates diverse solutions with the difficulty-aware rejection sampling. Recent studies (Shao et al., 2024; Hu et al., 2025; Zeng et al., 2025; Luo et al., 2025b; Yu et al., 2025; Yue et al., 2025) have explored reinforcement learning in complex reasoning tasks and have acquired great achievements. Inspired by recent successes in reinforcement learning for complex reasoning tasks, we propose TREERPO, an innovative reinforcement learning method that leverages tree sampling to further enhance LLM reasoning ability.

## 2.2 REINFORCEMENT LEARNING WITH LLMS

Reinforcement Learning from Human Feedback (RLHF; Ouyang et al. 2022) has been widely used in LLM alignments. Direct Preference Optimization (DPO; Rafailov et al. 2023) is further proposed to simplify the training pipeline of RLHF, which directly uses pair-wise preference data for model optimization. Recent studies (OpenAI, 2024; Guo et al., 2025; XAI, 2024; DeepMind, 2024; Qwen, 2024; Team et al., 2025) have shown that reinforcement learning can significantly improve the reasoning ability of models. This type of work can roughly be divided into two categories:

1) *Reward model-based*: There are two primary types of reward models: the Outcome Reward Model (ORM) and the Process Reward Model (PRM). Prior effort (Lightman et al., 2023) suggests that PRM outperforms ORM due to the fine-grained step-by-step reward signals. Math-Shepherd (Wang et al., 2024) trains a PRM by estimating the potential for a given reasoning step. However, training a reward model requires extensive, high-quality annotated data, especially for PRMs. This hinders the scaling of reward models in the field of complex reasoning.

2) *Reward model-free*: DPO is one of these methods, but it requires the elaborate construction of pairwise data for training. Step-DPO (Lai et al., 2024) constructs a pipeline to generate pair-wise step-level data and surpasses the performance of DPO. The other line of research (Shao et al., 2024; Hu et al., 2025; Zeng et al., 2025; Luo et al., 2025b) has shown that verification functions are effective in improving the reasoning capabilities of LLMs. They avoid the need for reward models, offering a simple yet effective approach. The typical methods are GRPO (Shao et al., 2024) and its variants DAPO (Yu et al., 2025) and VAPO (Yue et al., 2025). However, rule-based reward is similar to ORM, providing trajectory-level reward signals rather than fine-grained process reward signals. Unlike existing efforts, TREERPO achieves fine-grained, dense reward signals without relying on a separate reward model. TREERPO can offer a more scalable solution for obtaining dense reward signals in complex reasoning tasks.

## 3 TREERPO: METHODOLOGY

In this section, we elaborate on the proposed TREERPO. First, we present tree sampling in Section 3.1, which is designed to construct step step-level group to enhance long-chain reasoning abilities with GRPO. Next, in Section 3.2, we introduced the pruning strategy to improve the sampling and training efficiency in TREERPO. In Section 3.3, we discuss the numerical influence of standardized binary rewards and continuous rewards on advantage computation and propose a new advantage computation method for continuous rewards.

### 3.1 TREE SAMPLING

While GRPO has been proven to be effective and suitable for scaling in complex reasoning tasks with verifiable reward, it only provides the trajectory-level reward by evaluating the final answer of the generated sequences. Instead, to provide step-level reward estimation without using a reward model, we designed tree sampling.

Given an input question $q$, the language model generates an $N$-ary tree through iterative sampling, governed by the following constraints:

- **Branching Factor**: At each decoding step, the model samples $N$ candidate continuations, expanding $N$ new branches from the current node.

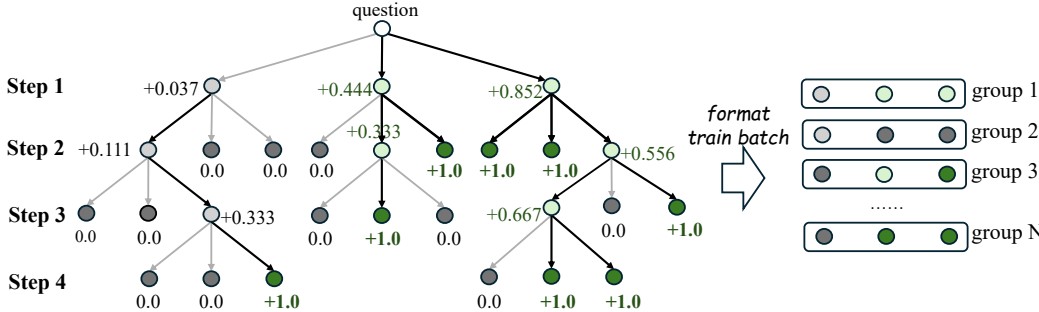

Figure 2: The sampling process of our TREERPO. TREERPO starts from the question, sampling $N$ nodes at each step until generation is completed or the maximum depth limit $D$ is reached. Then, a verifiable reward function is used to evaluate all leaf nodes and then back-propagates the rewards to their parent nodes, thereby obtaining intermediate step rewards, which achieves process reward signaling. We traverse each node and aggregate all children steps of a node into a group to compute advantages, which are finally formatted into the training batch.

- **Depth Limit**: The tree expansion terminates when any path reaches the maximum depth $D$, ensuring tractability.

- **Step Segmentation**: We directly divide the steps according to the token length. Each step produces at most $L_{\text{step}}$ tokens per branch. Generation halts for a branch if a stop token is generated, or the branch violates reaches depth limit. A more precise step division method is our future work.

As shown in Figure 2, the tree's reward computation follows a bottom-up recursive expectation scheme, where:

- **Leaf Evaluation**: For each leaf node $v_{\text{leaf}}$, the verification function $\phi$ takes the *entire path* $P = [v_{\text{root}}, \ldots, v_{\text{leaf}}]$ as input and computes the reward:

$$r_{\text{leaf}} = \phi(P) = \phi\big([v_{\text{root}}, \ldots, v_{\text{leaf}}]\big),$$

- **Parent Propagation**: Non-leaf nodes aggregate rewards from their children:

$$r_{\text{node}} = \mathbb{E}_{c \in \text{Children}(v_{\text{node}})}\big[r_c\big].$$

This propagates bottom-up, weighting all viable completion paths.

In conclusion, our tree sampling framework estimates the reward of each step as its potential to deduce the correct final answer.

## 3.2 DATA PRUNING

Similar to the Dynamic Sampling strategy of DAPO, we filter out the samples to keep all data samples in the training batch with effective gradients.

In the data construction pipeline of TREERPO, a **group** $\mathcal{G}$ is formally defined as the set of child nodes $c_1, \ldots, c_n$ originating from a common parent node $p$, as illustrated in Figure 2. Adopting a strategy analogous to the dynamic sampling approach in DAPO, we perform group-level filtering based on reward distribution characteristics.

$$\Delta R_{\mathcal{G}} = \max_{c_i \in \mathcal{G}} R(c_i) - \min_{c_j \in \mathcal{G}} R(c_j) \tag{1}$$

where $R(c_i)$ denotes the reward associated with child node $c_i$. We introduce a variance threshold $\tau$ such that a group $\mathcal{G}$ is included in the training batch $\mathcal{B}$ if and only if:

$$\mathcal{G} \in \mathcal{B} \iff \Delta R_{\mathcal{G}} > \tau \tag{2}$$

The threshold $\tau$ operates as a hyperparameter controlling the trade-off between sample diversity and learning signal strength in the batch construction process.

This data selection criterion ensures all samples in the batch with effective gradients and improves the efficiency of the training process.

### 3.3 ADVANTAGE COMPUTATION

In the vanilla GRPO framework, the advantage estimation is derived by normalizing binary rewards:

$$\hat{A}_{i,t} = \frac{r_i - \text{mean}(\{R_i\}_{i=1}^G)}{\text{std}(\{R_i\}_{i=1}^G)}. \tag{3}$$

However, when applied to continuous rewards, this approach introduces significant bias. For instance, two reward sequences, $\mathbf{R}_1 = [0, 0, 1, 1]$ and $\mathbf{R}_2 = [0.49, 0.49, 0.51, 0.51]$, produce identical normalized advantages $[-1, -1, 1, 1]$, despite their distinct reward distributions. While $\mathbf{R}_1$ exhibits a clear bimodal separation, $\mathbf{R}_2$ contains only minor variations (a maximal difference of $0.02$). This indicates that standard normalization fails to properly scale advantages for continuous rewards, leading to misleading policy updates.

To mitigate this bias, we propose an alternative advantage computation that preserves the statistical properties of binary reward normalization while accommodating continuous rewards. Instead of computing the empirical variance from $\mathbf{R}$, we define the normalization factor as $\sigma = \mu(1 - \mu)$, where $\mu$ is the mean reward. This formulation maintains consistency with the variance of Bernoulli-distributed rewards ($\text{Var}[R] = \mu(1 - \mu)$) while generalizing to continuous settings.

For a given reward sequence $\mathbf{R} = [R_1, R_2, \ldots, R_n]$, the advantage is computed as:

$$\mu = \frac{1}{n}\sum_{i=1}^n R_i \qquad \sigma = \mu(1 - \mu)$$
$$A_i = \frac{R_i - \mu}{\sigma} \tag{4}$$

By fixing the variance term $\sigma$ to $\mu(1 - \mu)$, we ensure that advantage values remain interpretable and stable, avoiding the overamplification of small differences in continuous rewards. This approach bridges the gap between binary and continuous reward normalization while maintaining the original scaling behavior of GRPO.

### 3.4 OBJECTIVE OF TREERPO

We adopt the clipped objective of GRPO, together with a directly imposed KL penalty term: Additionally, the KL-regularization between current policy $\pi_\theta$ and the reference policy $\pi_{\text{ref}}$ is directly added to the loss function:

$$\mathcal{J}_{\text{TreeRPO}}(\theta) = \mathbb{E}_{(q \sim \mathcal{D}, \{o_i\}_{i=1}^G \sim \pi_{\theta_{\text{old}}}(q))}$$
$$\left[ \frac{1}{G}\sum_{i=1}^G \frac{1}{|o_i|}\sum_{t=1}^{|o_i|} \left( \min\left(r_{i,t}(\theta)\hat{A}_{i,t}, \text{clip}\left(r_{i,t}(\theta), 1-\varepsilon, 1+\varepsilon\right)\hat{A}_{i,t}\right) - \beta D_{\text{KL}}(\pi_\theta||\pi_{\text{ref}}) \right) \right], \tag{5}$$

where

$$r_{i,t}(\theta) = \frac{\pi_\theta(o_{i,t} \mid q, o_{i,<t})}{\pi_{\theta_{\text{old}}}(o_{i,t} \mid q, o_{i,<t})}. \tag{6}$$

## 4 EXPERIMENTS

**Datasets.** We conduct the evaluation of our experiments using 4 widely used mathematical reasoning benchmarks: MATH-500 (Lightman et al., 2023), OlympiadBench (He et al., 2024), MinervaMath (Lewkowycz et al., 2022), and AIME24. Among them, Math-500 are 500 items screened out from the original MATH test split. The subset consists of 500 representative problems, and the

evaluation produces similar results to the full-set evaluation. In the training scenario, we use the training split of MATH dataset, which contains 7.5k high-quality training samples. In the future, we will extend the experiment to the DeepScaler (Luo et al., 2025b) training data, which is a more challenging dataset for mathematical reasoning.

**Parameter Setting.** Our experiments are based on Qwen2.5-Math series language models (Yang et al., 2024a). In the evaluation procedure, we set the temperature as 0.6 to sample 8 candidate responses for each question. In the reinforcement learning training procedure, we set the temperature as 0.6 and roll out 8 responses for each question. The learning rate is 1e-6 for both GRPO and TREERPO. The coefficients for KL divergence and entropy loss are $\beta = 0.001$ and $\alpha = -0.001$, respectively. For GRPO, the training batch size is 128 and the mini-batch size is 64. For our TREERPO, the training batch size is 128. Since the training data size of each step of TREERPO is floating, the size of our mini-batch is obtained as half of the training data size. By default, the maximum prompt length is 512, and the maximum response length is 1152 for GRPO. For TREERPO, the maximum prompt length is 512, the maximum step length $L_{step}$ is 384, the maximum depth $D$ of tree sampling is set as 3, and the $N$-ary is set as 8. For better efficiency, we set the data pruning coefficient $\tau$ to 0.1 as described in Sec. 3.2.

**Implementation Details.** We follow the rllm (Luo et al., 2025a;b) framework which is derived from the verl (Sheng et al., 2024) pipeline. Both rllm and verl integrate the vllm (Kwon et al., 2023) framework for efficient inference of models. All of our experiments are conducted on A800 GPUs. At present, the LLM of our experiment is the Qwen2.5-Math series. Due to the limitations of time and computation resources, we have only reported the 1.5b model. We plan to conduct experiments on 7b and 32b as soon as possible

**Metrics.** We use the same verification function in rllm to evaluate the performance of LLMs. Compared with other repositories, the reward function implemented by rllm is more complete and systematic. For the test results, the accuracy rate we report is **pass@1(avg@8)** performance for all tested benchmarks.

Table 1: Overall performance of *Pass@1* (*Avg@16*) performance of Qwen2.5-Math series.

| **Method** | AIME24 | MATH500 | Olympiad | Minerva | Macro Accuracy |
|---|---|---|---|---|---|
| *Qwen2.5-Math-1.5B as the Base Model* | | | | | |
| GRPO Baseline | 13.8 | 67.9 | 28.5 | 20.5 | 32.7 |
| TreeRPO | 16.8 (↑**+3.0**) | 70.7 (↑**+2.8**) | 30.9 (↑**+2.6**) | 24.0 (↑**+3.5**) | 35.6 (↑**+2.9**) |
| *Qwen2.5-Math-7B as the Base Model* | | | | | |
| GRPO Baseline | 26.7 | 74.3 | 34.7 | 27.1 | 40.7 |
| TreeRPO | 26.7 | 75.5 (↑**+1.2**) | 35.4 (↑**+0.7**) | 28.1 (↑**+1.0**) | 41.4 (↑**+0.7**) |

## 4.1 MAIN RESULTS

We show the performance comparison of GRPO baseline and our TreeRPO on Qwen2.5-Math-1.5/7B in four selected benchamrk: AIME24, MATH-500, Olympiad Benchamrk, and Minerva Math. As illustrated in Table 1, for Qwen2.5-Math-1.5B, the Macro Accuracy has improved by 2.9%. Furthermore, we consider that the reason why the improvement and repetition of Qwen2.5-Math-7B is not as significant as that of Qwen2.5-Math-1.5B lies in the fact that the MATH training data for Qwen2.5-Math-7B is too simple, resulting in the improvement of the algorithm not being significantly reflected. In generall, our **TreeRPO** has achieved a consistency improvement compared to GRPO baseline.

**TREERPO demonstrates significant performance improvements.** We conduct TREERPO and GRPO on Qwen2.5-Math-1.5b model with the training split of the MATH dataset, and conduct the evaluation on four selected benchmarks: Math-500, MinervaMath, OlympiadBench, and AIME. As

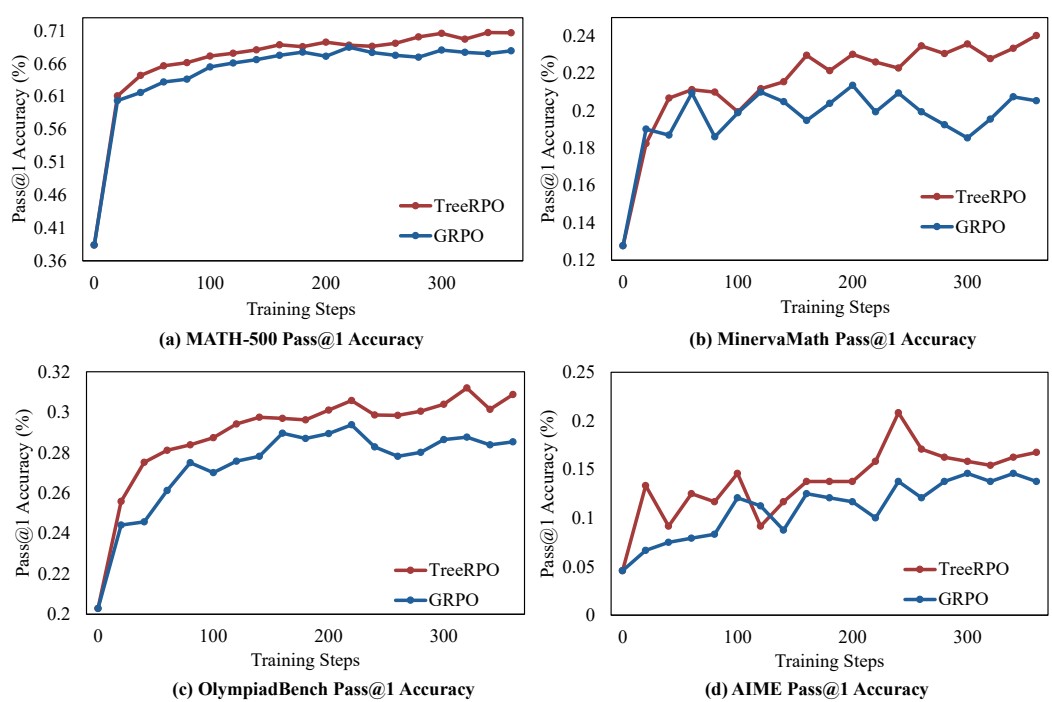

Figure 3: Performance comparison of our TREERPO and GRPO on the four selected benchmarks: Math-500, MinervaMath, OlympiadBench, and AIME. The experiments are conducted with Qwen2.5-Math-1.5b, an LLM pretrained with a large amount of mathematical corpus.

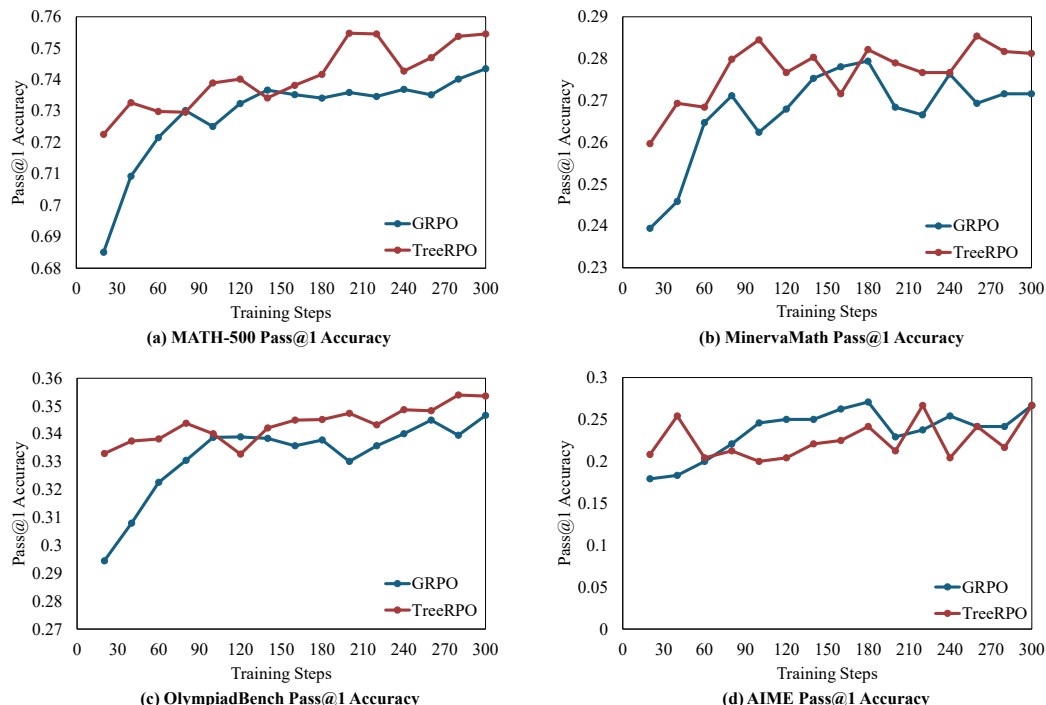

Figure 4: Performance comparison of our TREERPO and GRPO on the four selected benchmarks: Math-500, MinervaMath, OlympiadBench, and AIME. The experiments are conducted with Qwen2.5-Math-1.5b.

shown in Figure 3, our TREERPO outperform GRPO on all of the tested benchmarks. We further show the dynmaic results of Qwen2.5-Math-7B in Figure 4. After training 300 steps for Qwen2.5-Math-1.5B, our TREERPO outperforms GRPO by 2.7% on MATH-500, 3.5% on MinervaMath, 2.4% on OlympiadBench, and 3.0% on AIME, respectively. As illustrated in Figure 1, TREERPO outperforms the overall performance of GRPO by **2.9%**. In conclusion, TREERPO has demonstrated consistent superiority on multiple benchmarks.

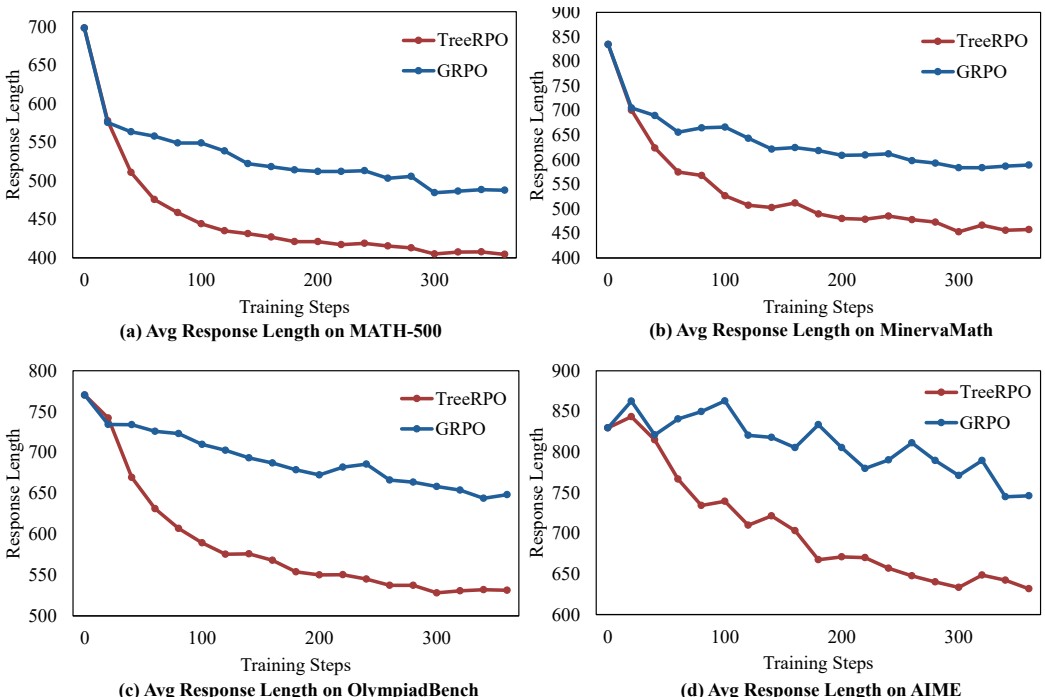

Figure 5: Response Length comparison of our TREERPO and GRPO on the four selected benchmarks: Math-500, MinervaMath, OlympiadBench, and AIME. The experiments is conducted with Qwen2.5-Math-1.5b

**TREERPO demonstrates efficiency advantage in token usage.** We conduct TREERPO and GRPO on the Qwen2.5-Math-1.5b model with the training split of the MATH dataset, and compute the average response length on four selected benchmarks: Math-500, MinervaMath, Olympiad-Bench, and AIME. As illustrated in Figure 5, compared to GRPO, our TREERPO achieves a 17.1% reduction in token usage on MATH, 22.3% on MinervaMath, 18.0% on OlympiadBench, and 15.3% on AIME. On average, TREERPO demonstrates a **18.1%** decrease in token usage across the four benchmarks compared to GRPO, showcasing its superior efficiency. TreeRPO not only demonstrates an advantage in token efficiency on Qwen2.5-Math-1.5B, but also shows an efficiency advantage on Qwen2.5-Math-7B, with an average token length that is also shorter than the GRPO baseline. We show the response case of a simple question in Figure 7. It can be seen that in this simple case, TREERPO's response is more concise

**The performance of TREERPO under different hyperparameters.** In the experiments, we conduct experimental analyses using different batch sizes, and the results are shown in Figure 6. For GRPO and TREERPO, the batch size $bsz = 16/128$ has very little influence on the final performance. Our TREERPO significantly outperforms GRPO in both two Settings. This fully demonstrates that our TreeRPO algorithm significantly outperforms the GRPO-baseline across different hyperparameters.

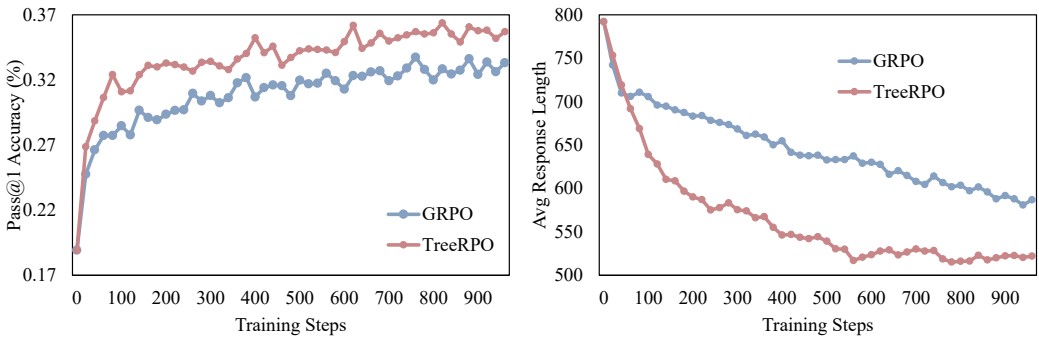

Figure 6: Comparison of TREERPO and GRPO with $bsz = 16$. The pass@1 accuracy and the response length are calculated by taking the average on the four tested benchmarks.

---

**Question**: Solve $\frac{1}{x-5} > 0$. Enter your answer using interval notation.

---

**TreeRPO**:

To solve the inequality $\frac{1}{x-5} > 0$, we need to determine when the expression $\frac{1}{x-5}$ is positive. The fraction $\frac{1}{x-5}$ is positive when the denominator $x - 5$ is positive. Therefore, we need to solve the inequality $x - 5 > 0$.

Adding 5 to both sides of the inequality, we get:
$$x > 5$$
So, the solution to the inequality $\frac{1}{x-5} > 0$ is $(5, +\infty)$.

The final answer in interval notation is:
\boxed{(5, +\infty)}.

GRPO:

To solve the inequality $\frac{1}{x-5} > 0$, we need to determine the values of $x$ for which the expression $\frac{1}{x-5}$ is positive. The expression $\frac{1}{x-5}$ is positive when the denominator $x - 5$ is positive. Therefore, we need to solve the inequality $x - 5 > 0$.

Let's solve this step-by-step:

1. Start with the inequality:
$$x - 5 > 0$$
2. Add 5 to both sides of the inequality:
$$x > 5$$
So, the solution to the inequality $\frac{1}{x-5} > 0$ is $x > 5$. In interval notation, this is written as $(5, +\infty)$.

The final answer is: \boxed{(5, +\infty)}.

Figure 7: Responses of TREERPO and GRPO of a simple question.

## 5 CONCLUSION

In this paper, we propose TREERPO, which conducts tree sampling to construct step-level groups based on vanilla GRPO. TREERPO obtains the reward of the current step by estimating the reward of the subsequent sampling paths of the current step. This is a method that can obtain dense reward signals without the need for process reward models (PRMs). The experimental results show that TREERPO demonstrates both effectiveness and efficiency. In the future, we will continuously improve the algorithm based on the current version and expand the scale of LLM training.

## 6 REPRODUCIBILITY STATEMENT

To ensure the reproducibility of our research, we have meticulously assembled a comprehensive reproducibility package as part of our supplementary materials. This package is designed to enable the seamless replication of all experiments detailed in our paper. It encompasses anonymized source code that implements the proposed model and training procedures. Additionally, we have included precise configuration files and scripts that specify all hyperparameters and the training commands necessary to reproduce our results.

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

# APPENDIX

## A  THE USE OF LLMs

In the preparation of this paper, large language models (LLMs), specifically DeepSeek-V3.1 and Gemini 2.5, were used solely for the purpose of polishing the writing. The LLM was employed after the core intellectual content—including the central ideas, theoretical formulations, algorithm designs, experimental setups, and result analyses—had been fully developed by the authors. The model's assistance was limited to rephrasing sentences for improved clarity, fluency, and conciseness. All prompts provided to the LLM contained only the authors' original text and instructions for grammatical or stylistic improvement.

## B  FUTURE WORK AND LIMITATIONS

**Remove Redundant Steps.** Yuan et al. (2023) uses Rejection Sampling to collect correct reasoning paths for training LLMs. They find that the sampled redundant responses degrade the performance of LLMs. We consider that this phenomenon may also exist in RL. In vanilla GRPO, each response is treated equally, so responses with high similarity are repeatedly trained, which may cause performance disturbances. We believe that eliminating redundant rollouts can enhance performance while improving training efficiency through pruning.

**Precise Step Segmentation.** The step division of generated sequences in this article is implemented based on a specific token length. Give priority to exploring more precise step division methods.

- One solution to be implemented is to add the **step special token** and train the language model to segment different steps by itself.
- Sampling at the tokens where branch paths are more likely to be generated (Wang et al., 2025).

We believe that more precise step cutting will provide more accurate fine-grained reward signals and further enhance the model's performance.

**Scaling on Larger Model Sizes.** Due to the limitations of time and GPU resources, our experiment can only report the 1.5b model for the time being. The experimental results of larger-sized models, such as 7b and 32b, will be updated in the future.

**Engineering Efficiency Optimization of Tree Sampling.** Tree sampling is time-consuming, and the tree sampling strategy implemented in this paper is not optimized from the perspective of the KV cache. We believe that the engineering optimization of tree sampling will significantly improve the efficiency of the training procedure.

