# OpenReview forum: "TreeRPO: Tree Relative Policy Optimization"
_ICLR.cc/2026/Conference — ICLR 2026 Conference Withdrawn Submission_

### Official Review · Reviewer_g2qw · 2025-10-29

**Soundness:** 2
**Presentation:** 3
**Contribution:** 2
**Rating:** 2
**Confidence:** 3

**Summary:**

TREERPO addresses the crucial limitation of trajectory-level model-free RL methods like GRPO: dence rewards. It introduces a novel approach for fine-grained intermediate step reward estimations for training LLM in deterministic settings. It leverages tree sampling to construct step-level groups for estimation of step-level rewards, extending the group-relative policy optimization (GRPO) framework. The method improves reasoning accuracy on mathematical benchmarks, outperforming GRPO.

**Strengths:**

1. Reward model-free step-level reward estimation: TREERPO addresses the crucial limitation of trajectory-level RL: dense rewards.

2. Efficiency and performance gains: experimental results demonstrate that TREERPO significantly boosts Pass@1 accuracy by 2.9% over GRPO on multiple mathematics benchmarks and provide improved computational efficiency

**Weaknesses:**

1. There is no theoretical grounding provided for the proposed method: does it yield more effective estimates for the loss? Can we compare the variance of these estimates with those of GRPO? There should at least be some intuition, examples, formulas, or simulation studies. Since GRPO can be seen as a special case of the TreeRPO algorithmically (when the branching factor is 1), why does a greater branching factor lead to better results? To me, there is no clear intuition supporting this.

2. In addition to the tree sampling approach used to estimate rewards, the authors introduce a KL divergence term in the objective function (Section 3.4). Without ablation studies, it is unclear which of these two factors contributes most to the model improvements observed in experiments. Could you provide such details?

3. TREERPO does not appear to be the first method for dense step-level rewards in the model-free deterministic setting. What about "Exploiting Tree Structure for Credit Assignment in RL Training of LLMs" by Tran et al.? This paper was published two days before the ICLR deadline. Given the conference's high standards, a comparison between this submission and that paper is requested.

4. The empirical evidence seems unconvincing: there is evaluation on only one architecture, Qwen2.5-Math (with two model sizes), and improvements are smaller for the larger model. Why does the improvement diminish with the larger model? The explanation at lines 317-318 states: "the MATH training data for Qwen2.5-Math-7B is too simple." This sounds vague and unconvincing because (1) no references are provided, (2) how does this correspond with the larger Pass@1 for the 7B model? (3) More importantly, how is this expected to influence the difference in effectiveness between GRPO and TreeRPO in theory — or is it an orthogonal factor?

**Questions:**

See Weaknesses for explicit questions

---

### Official Review · Reviewer_1JLT · 2025-10-30

**Soundness:** 3
**Presentation:** 3
**Contribution:** 3
**Rating:** 4
**Confidence:** 3

**Summary:**

This paper introduces TreeRPO, which uses tree-based sampling to estimate step-level rewards without requiring an explicit reward model e.g. PRM. The method is an extension of GRPO, generating a tree at each decoding step to construct step-level groups. The contributions are as follows: introducing the first reward model-free method that provides dense reward signals through tree sampling, demonstrating significant performance improvements over GRPO, and showing improved token efficiency.

The strengths of the work are as follows. First, the paper touches on an important subject in which GRPO is a critical part of training today's LLMs. Second, the technical approach is generally sound, and the sampling approach is well-motivated. Finally, the paper is empirically strong and shows a large performance improvement on Qwen-2.5-Math-1.5B and outperforms GRPO.

The weaknesses are as follows. First, the sampling approach is quite computationally expensive. Although the approach is model free, it still requires a large cost for sampling. Second, the ablation studies could be improved e.g. with additional analysis on the branching factor or the depth of the tree. Finally, the model size tested is a bit small, and it's curious about how the method would do on larger model families.

**Strengths:**

The strengths of the work are as follows.
- First, the paper touches on an important subject in which GRPO is a critical part of training today's LLMs. - Second, the technical approach is generally sound, and the sampling approach is well-motivated.
- Finally, the paper is empirically strong and shows a large performance improvement on Qwen-2.5-Math-1.5B and outperforms GRPO.

**Weaknesses:**

The weaknesses are as follows.
- First, the sampling approach is quite computationally expensive. Although the approach is model free, it still requires a large cost for sampling.
- Second, the ablation studies could be improved e.g. with additional analysis on the branching factor or the depth of the tree.
- Finally, the model sizes (1.5B, 7B) tested is a bit small, and it's curious about how the method would do on larger model families. Additionally, the performance gain is not as great with the 7B model, which is a potential concern.

**Questions:**

Has TreeRPO been tested on non-math tasks?

---

### Official Review · Reviewer_vCgU · 2025-11-01

**Soundness:** 2
**Presentation:** 3
**Contribution:** 2
**Rating:** 2
**Confidence:** 4

**Summary:**

TreeRPO introduces a reinforcement-learning algorithm that supplies dense, step-level reward signals for fine-tuning LLMs on mathematical reasoning tasks without requiring a learned process-reward model.
The key technical idea is to (i) sample an N-ary tree of reasoning continuations up to a fixed depth, (ii) label every leaf with a verifiable outcome reward, and (iii) back-propagate expected returns bottom-up to obtain a pseudo-reward for every intermediate node.
These per-step returns are then used inside a GRPO-style group-relative policy objective.
On Qwen2.5-Math-1.5B the method improves Pass@1 from 19.0% → 35.5% on a 4-bench suite (MATH-500, OlympiadBench, Minerva, AIME24) while shortening average outputs by 18.1% relative to the GRPO baseline.

**Strengths:**

- Originality: First reward-model-free RL approach that delivers per-step (process-style) supervision via tree sampling rather than learned PRMs or Monte-Carlo roll-outs
- Empirical gains: Consistent +2–3% absolute accuracy improvements over a strong GRPO baseline on four maths benchmarks, with lower token cost
- Clarity of method: The recursive reward back-propagation and group-level filtering are easy to implement and clearly described.
- Well-scoped: Stays within verifiable-reward domains (math with known answers), avoiding annotation bottlenecks that plague PRM training.

**Weaknesses:**

Missing baselines
- No comparison with any PRM-based method (e.g., Math-Shepherd, Wang et al. 2024) or step-level RL (Step-DPO, Lai et al. 2024). Hence the claim “first to provide dense signals without a reward model” is not sufficient to establish superiority over existing PRM pipelines.

Statistical rigor
- Results are reported as single-run curves (Fig. 3) without standard deviations or confidence intervals. With only 500–1k test questions, variance can be high.

Ablations incomplete
- Tree depth D=3, branching N=8, and pruning τ=0.1 are fixed without ablation.
- Advantage re-normalisation  is motivated with a toy example but not ablated; it is unclear how much it contributes.

Limited scale and scope of experiments
- Main claims rely mostly on Qwen2.5-Math-1.5B; 7B results are limited and not systematically analyzed (sec. 4.1 mentions 7B but claims limited gains). The benefit may not hold at larger scales.

Step segmentation heuristic
- Steps are split by token length L_step=384 (sec. 3.1) which may break semantic boundaries and bias the per-step rewards.

**Questions:**

- What is the compute overhead (GPU-hours) of tree sampling relative to GRPO under the same sample budget?
- Did you try D=1 (i.e., GRPO with N roll-outs) and N=1 while keeping total samples fixed? This would isolate the tree back-prop contribution from mere sample diversity.
- Why were PRM-based baselines not included in the experiments?
- How sensitive are results to the pruning threshold τ? A quick sweep plot would suffice.
- Do gains persist when response length is forced to be identical (e.g., via length penalty)? This checks whether accuracy lift is partly due to shorter, less noisy outputs.
- What is the exact verification function implementation (rationale and code) used to evaluate leaf nodes? Is it identical for TreeRPO and GRPO?
- Does TreeRPO still outperform GRPO on larger models (7B, 32B) or broader reasoning tasks beyond math? If not tested, can you provide preliminary results to underdstand the generality of the approach?

---

### Note · Authors · 2025-11-21

I have read and agree with the venue's withdrawal policy on behalf of myself and my co-authors.